**Data Availability Statement:** All data are in the manuscript and/or supporting information files

# Bacterial aetiology, antimicrobial susceptibility patterns, and factors associated with urinary tract infection among under-five children at primary health facility, North-Western Tanzania

**Roza Ernest** [1,2]*, **Nsiande Lema**[1], **Sued Yassin**[3], **Agricola Joachim**[4], **Mtebe Majigo**[4]

**1** Tanzania Field Epidemiology and Laboratory Training Program, Dar es Salaam, Tanzania, **2** Department of Epidemiology and Biostatistics, Muhimbili University of Health and Allied Sciences, Dar es Salaam, Tanzania, **3** Research, Training, and Consultancy Unit, Chato Zonal Referral Hospital, Geita, Tanzania, **4** Department of Microbiology and Immunology, Muhimbili University of Health and Allied Sciences, Dar es Salaam, Tanzania

* roza.ernest@yahoo.com

## Abstract

### Background

Urinary tract infections (UTI) are common in under-five children, with significant consequences leading to bacteremia, dehydration, kidney scarring, and renal failure. The incidence of UTI varies with patients' demographics and geographic location. Limited studies have addressed UTI issues, particularly in children. We determined the proportion of UTI, bacterial aetiology, and antimicrobial susceptibility patterns and associated factors among under-five children at the district hospital between March and April 2023

### Methods

We conducted a cross-sectional study using a convenient non-probability sampling technique to collect urine samples from participants with signs and symptoms of UTI. Written informed consent was obtained from parents or guardians. We collected Participants' information using a pretested structured questionnaire. Urine samples were processed at the Regional Referral Hospital. All analyses were conducted using STATA version 15.0. We determined the factors associated with UTI using a modified Poisson model multivariable analysis of the modified Poisson model. The results were presented as a prevalence ratio and 95% confidence interval. The level of significance was specified at 0.05.

### Result

The study recruited 368 under-five children; 194 (52.7%) were males, and the median age (interquartile range) was 24 (13–36) months. Of all, 28.8% (95% CI-24.3–33.6) had culture-confirmed UTI. One hundred and six pathogens were isolated, the majority being *Escherichia coli (E. coli)*, 37 (34.9%), and *Staphylococcus aureus (S. aureus)*, 26 (24.5%). The

**Funding:** This work is supported by the United States President's Emergency Plan for AIDS Relief (PEPFAR) through the Centre of Excellence in Health Monitoring and Evaluation, Mzumbe University under the U.S Centers for Disease Control and Prevention (CDC), Project Cooperative Agreement No: NU2GGH002292 which supports Tanzania Field Epidemiology and Laboratory Training Program (TFELTP), however, the funder did not contribute to the study's design, data collection, analysis, interpretation, or manuscript writing.

**Competing interests:** The authors of this research have explicitly stated that there are no existing conflicts of interest related to the publication of their work.

**Abbreviations:** ATCC, American Type Culture Collection; CFU, Colony-Forming Unit; CLSI, Clinical and Laboratory Standards Institute; ESBL, Extended Spectrum Beta Lactamase; MDR, Multi-Drug Resistant; MRSA, Methicillin-Resistant *Staphylococcus Aureus*; UTI, Urinary Tract Infection; WHO, World Health Organization.

susceptibility of *E. coli* to cefepime, piperacillin-tazobactam, nitrofurantoin, and meropenem ranged from 81.1% to 97.3%. *S. aureus* was most susceptible to nitrofurantoin (96.2%) and ciprofloxacin (92.3%). Multidrug resistance was observed in 33.0% of isolates. The proportion of Methicillin-resistant *S. aureus* and extended-spectrum beta-lactamases was 23.1% and 25%, respectively. UTI was observed more in patients presenting with vomiting, dysuria, and abdominal pain, patients below 24 months of age, nappy users, and uncircumcised males.

## Conclusion

Our study found a relatively high proportion of UTI among under-five children associated with vomiting, dysuria, abdominal pain, nappy use, and uncircumcision in males. The pathogens were least susceptible to (trimethoprim-sulfamethoxazole, gentamycin, ampicillin, and penicillin) the commonly used antibiotic. We advocate a thorough clinical analysis to detect the predictors of UTI and a periodic review of empirical treatment of UTI based on the antibiotic susceptibility pattern.

## Introduction

Urinary tract infections (UTI) are common in under-five children, with significant consequences leading to bacteremia, dehydration, kidney scarring, and renal failure [1]. Early diagnosis is essential to preserve and maintain the proper functioning of the kidneys and prevent further decline in their function [2,3]. UTIs have been reported among the most typical sites of infection in the community, and the incidence of infection varies according to age, gender, and health condition [4]. Some studies have reported the prevalence of UTI in under-five children ranging from 7.4% to 43.6% [1,5,6].

UTI in children is primarily caused by bacteria, but other pathogens such as viruses, fungi, and parasites can also be responsible. The most common bacteria associated with UTI in children include gram-negative pathogens such as *Escherichia coli (E. coli)*, *Klebsiella*, *Proteus*, *Enterobacter*, *Pseudomonas*, and *Serratia*. Gram-positive pathogens associated with UTI in children include group B *streptococcus*, *enterococcus* spp, and *Staphylococcus aureus (S. aureus)*. In addition to bacterial causes, non-bacterial causes include viruses causing viral haemorrhagic cystitis and Candida infection, particularly in individuals with compromised immune systems [6–8]. The bacteria causing UTIs have varied antibiotic susceptibility patterns and are most resistant to commonly used antibiotics [6,9].

The risk factors for acquiring UTI in under-five children are multiple and differ with geographical locations, settings, and seasons [10]. The commonly reported factors include female gender, poor hygiene, poverty, malnutrition, and recurrent infections [11]. Other predisposing factors include lower age, uncircumcision in males, genitourinary anomalies, indwelling urinary catheters, and previous antibiotic use [12,13]. Urine culture is rarely performed as part of patient management in rural areas with inadequate diagnostic resources. Treatment is empirical, which could encourage the development and spread of antimicrobial-resistant bacteria [14]. Recent research has shown a high prevalence of resistance to commonly used antibiotics [14–16]. Antimicrobial resistance is also prevalent in Tanzania, with reports indicating multi-drug resistance among urinary pathogens associated with UTIs [17].

Extended-spectrum beta-lactamases (ESBL) producers contribute a high resistance rate to beta-lactam antibiotics and are more common in hospitals than in community settings [17,18]. ESBL-producing bacteria prevalence of infection or colonisation among children under five years ranges from 17% to 35% [18–21]. Studies in Tanzania reported that ESBL producers' prevalence ranges from 18% to 45%; Carbapenems were reported as the drugs to treat ESBL-producing bacteria [17,22]. However, they are expensive and not widely available in Tanzania.

Therefore, increasing knowledge of the aetiologic profile and antimicrobial susceptibility patterns of pathogens associated with UTI in under-fives is essential in medical practices. Understanding the associated factors also helps identify at-risk groups and institute preventive measures. This research paper explored the etiologic profile, antimicrobial susceptibility pattern, and factors associated with UTI among under-five children in Magu District Hospital, Mwanza-Tanzania.

## Materials and methods

### Study design, setting, and population

We conducted a cross-sectional study between 8th March and 14th April 2023 at Magu District Hospital in Mwanza Region, North-Western Tanzania. Magu district is located between 33˚ and 34˚ East of Greenwich and between 2˚10′ and 2˚50′ South of the Equator. Its total area is 3075 km2, of which Lake Victoria waters cover 1725 km2 (56.1%). The average annual rainfall is between 700 and 1000 mm. As of 2012, the population of Magu district was 299,759 people. Magu District Hospital is a referral hospital in the district. Over 40% of the patient population seeking services is under-fives children. The laboratory procedures were conducted at Sekou Toure Regional Referral Hospital Laboratory.

The study involved under-five children seeking healthcare services. We recruited children whom the attending clinician suspected to have UTI based on signs and symptoms such as fever of 38 degrees Celsius or above, pain or burning when urinating, frequent urination, discoloured urine, lower abdominal pain, or vomiting. The study did not include children who used catheters within 72 hours because catheter insertion may directly inoculate microorganisms into the urinary bladder [23].

### Sample size and sampling technique

We estimated the minimum sample size of 355 under-five children using the Kish Leslie formula, considering the 11.4% previously reported prevalence of UTI among under-five children [24]. We used a non-probability convenient sampling technique, in which all febrile under-five children attending Magu district hospital were recruited in the study after obtaining consent from their parents/guardians.

### Data collection

The participants' socio-demographic (age, sex, and place of residence), clinical characteristics (vomiting, dysuria, body temperature, abdominal pain, and chronic disease), and risk factors for UTI (nutrition status, history of recurrent UTI, male circumcision, duration of fever, and nappy use) were collected using a structured and pre-tested questionnaire. We measured axillary temperature and body weight using a digital thermometer and digital weighing scale, respectively. The children's nutrition status was determined by calculating their weight-for-age z-scores (WAZ), representing how many standard deviations their weight is from the median weight of a healthy reference population of the same age and sex. a WAZ below -3

standard deviations was used to indicate malnutrition, while a WAZ between -2 and +2 standard deviations indicated a normal nutritional status (well-nourished) [25].

## Sample collection

The study participant's parents or guardians assisted with the urine collection after receiving adequate instruction. Approximately 10ml of urine specimens were collected into a sterile, wide-mouth, leak-proof, screw-capped plastic container pre-labelled with the participant's identification number, age, date, and time. The specimen was transported daily to Sekou Toure Regional Referral Hospital laboratory for processing within two hours and kept in a cool box. Immediate inoculation of samples was done on arrival at the laboratory. If a delay was inevitable, specimens were refrigerated at 4˚C not more than 12 hours.

## Definition of terms

- **Multidrug resistance** (MDR) was defined as resistance to at least one antimicrobial agent in three or more antimicrobial categories.

- **Methicillin-resistant *S. aureus*** was defined as an isolate of *S. aureus* with an inhibition zone of less than or equal to 21mm diameter around the 30μg cefoxitin disk on antimicrobial susceptibility disk diffusion method.

- **Extended-spectrum beta-lactamase producers** were defined as gram-negative bacteria that showed an increase in the inhibition zone towards the amoxicillin-clavulanic acid disk in either the cefotaxime (30μg) disk or the ceftazidime (30μg) disk placed in a straight line 20 mm apart on the Muller Hinton plate, with amoxicillin-clavulanic acid at the centre.

- **Urinary tract infection** was determined when bacterial growth of $\geq 10^5$ colony-forming units (CFU)/ml was observed in urine culture.

## Urine culture

We performed a quantitative urine culture to determine the significant bacteriuria. Urine samples were inoculated into Cysteine Lactose-Electrolyte Deficient (CLED) (Sigma Alorich, India) agar plates using a calibrated wire loop (0.001mL). Culture plates were incubated under aerobic conditions at 37˚C for 24 hours. Bacterial growth of $\geq 10^5$ colony-forming unit (CFU)/mL of urine was considered significant bacteriuria. Specimens yielding fewer than $10^5$ CFU/mL were considered insignificant or potentially contaminated. In instances of mixed bacterial growth, subculturing was conducted to isolate each organism separately.

## Bacteria identification

We used conventional methods to identify isolated bacteria, including colony characteristics, gram staining reaction, and biochemical testing. Sulphur Indole Motility (SIM) (Himedia, India), Kligler Iron Agar test (Oxoid Ltd, UK), urease test (Oxoid Ltd, UK), and citrate utilisation test (Oxoid Ltd, UK) were used to identify gram-negative bacteria. The oxidase test (Himedia, India) was used to differentiate *Pseudomonas* species from Enterobacterales. Catalase and coagulase tests were used to identify gram-positive bacteria. We differentiated *S. aureus* from other *Staphylococcus* and *Micrococcus* species by coagulase tests.

## Antimicrobial susceptibility testing

Antimicrobial susceptibility testing was performed using the Kirby-Bauer disk diffusion method, adhering to Clinical and Laboratory Standards Institute (CLSI) guidelines [26]. The selection of antibacterial disks was based on CLSI 2022 recommendations and availability. The bacterial inoculum, equivalent to 0.5 McFarland standards, was uniformly spread onto Mueller–Hinton agar plates (Himedia India) using a sterile cotton swab applicator. Antibiotic disks for gram-positive bacteria were penicillin G (10 $\mu$g), nitrofurantoin (300 $\mu$g), ciprofloxacin (5$\mu$g), trimethoprim-sulfamethoxazole (1.25/23.75 $\mu$g), gentamicin (10 $\mu$g), cefoxitin (30 $\mu$g), and clindamycin (30 $\mu$g) (Himedia, India).

The antimicrobial disks for Enterobacteriaceae included piperacillin-tazobactam (100/10 $\mu$g), ampicillin (10 $\mu$g), nitrofurantoin (300 $\mu$g), amoxicillin-clavulanate (20/10 $\mu$g), gentamicin (10 $\mu$g), cefotaxime (30 $\mu$g), ciprofloxacin (30 $\mu$g), meropenem (10 $\mu$g), piperacillin (100 $\mu$g), cefepime (30 $\mu$g), amikacin (30 $\mu$g), trimethoprim-sulfamethoxazole (1.25/23.75 $\mu$g), ceftazidime (30 $\mu$g), and ceftriaxone (30 $\mu$g) (Himedia, India). Antimicrobial disks for *Pseudomonas* spp and *Acinetobacter* were gentamicin (10 $\mu$g), meropenem (10 $\mu$g), ceftazidime (30 $\mu$g), piperacillin-tazobactam (100/10 $\mu$g), ciprofloxacin (5 $\mu$g), and amikacin (30 $\mu$g) (Himedia, India).

## Extended-spectrum beta-lactamase detection

Gram-negative bacteria producing ESBL was first suspected from the result of antimicrobial susceptibility testing based on the diameter of the zone of inhibition for the three third-generation cephalosporins: cefotaxime (30$\mu$g), ceftriaxone (30$\mu$g) and ceftazidime (30$\mu$g). The following breakpoints were used to suspect ESBL production: ceftriaxone $\leq$ 25mm, ceftazidime $\leq$ 22mm, and cefotaxime $\leq$ 27mm.

We confirmed ESBL production using the Double Disk Synergy method [27]. Ceftazidime (30$\mu$g), cefotaxime (30$\mu$g), and amoxicillin-clavulanic acid (20/10 $\mu$g) disks (Himedia, India) were placed in a straight line 20 mm apart on the plate, with amoxicillin-clavulanic acid in the centre. An increase in the inhibition zone towards the amoxicillin-clavulanic acid disk in either the cefotaxime disk or ceftazidime disk was a positive result for ESBL enzyme production; the isolate was reported as an ESBL producer.

## Detection of methicillin-resistant *Staphylococcus aureus*

Methicillin-resistant *S. aureus* (MRSA) was phenotypically determined using a cefoxitin (30 $\mu$g) disk (Oxoid Ltd, UK) per CLSI guidelines [26]. A cefoxitin disk was applied and incubated at 37˚C for 18 to 24 hours. A less or equal to 21 mm inhibition zone diameter around the cefoxitin disk indicated MRSA.

## Quality assurance

Media and reagents were checked by verifying the expiration date, storage conditions, and visual inspection of any signs of contamination, such as discolouration, turbidity, or particulate matter. The media quality was assessed using sterility and performance tests using reference strains. Reference strains for culture and susceptibility testing included *Pseudomonas aeruginosa* (*P. aeruginosa*) (ATCC 27853), *S. aureus* (ATCC 25923), and *E. coli* (ATCC 25922). Additionally, *E. coli* (ATCC 25922) and *K. pneumonia* (ATCC 700603) ESBL-positive strains were used to control ESBL production. *S. aureus* (ATCC 43300) was used as a positive control for MRSA.

## Data analysis

Data analysis was done using STATA version 15 StataCorp.2017. Frequency and proportions were used to summarise the characteristics of study populations for categorical variables, whereas median and interquartile range were used to summarise continuous variables. Pearson's chi-square test was used to compare a categorical variable to the status of UTI. The modified Poisson model was performed to determine factors associated with UTI. The multivariable analysis involved variables with a p-value ≤ 0.2 to determine the adjusted prevalence ratio and 95% confidence intervals. A p-value < 0.05 was used in both models to acknowledge statistical significance.

## Ethical approval

The study obtained ethical clearance number MUHAS-REC-12-2022-1484 from the Muhimbili University of Health and Allied Sciences Senate Research Committee and Publications Committee. In addition, we got permission to collect data from the offices of the regional and district medical officers. Parents or guardians were informed about the study objectives and consented on behalf of their children before inclusion in the study. Results of urine culture and antimicrobial susceptibility tests were communicated timely to the attending clinician for patient management. Participant identification numbers were used in the study instead of names to ensure confidentiality.

## Results

### Socio-demographic and clinical characteristics

A total of 368 under-five children were enrolled in the study. Males counted for more than half of the participants, 194 (52.7%). The study participants' median age (interquartile range) was 24 (13–36) months. Most participants, 296 (80.4%), lived in rural areas. Malnutrition was observed in 27 (7.3%) and 29 (14.9%) males were already circumcised. Most participants, 201 (54.6%), had a fever for 3 to 7 days, and the majority, 348 (94.6%), had a 38–39˚C temperature. Abdominal pain was observed in 153 (41.6%), while 61 (16.6%) had dysuria. Vomiting was observed in 84 (22.8%) participants, and 24 (6.5%) had a history of recurrent UTI (Table 1).

### The proportion of UTI among under-five children

UTI was detected in 28.8% (106/368, 95% CI -24.3–33.6), more in children below 24 months, 53 (35.1%), than in children aged 24 months and above, 53 (24.4%) ($p$ = 0.026). UTI was significantly higher in children with dysuria, 30 (49.2%) compared to those without dysuria, 76 (24.8%) ($p$ < 0.001). The proportion of UTI was higher among children with a fever of 3–7 days 68 (33.8%) compared to those with a fever of ≤ 2 days 33 (21.3%) and greater than 7 days 5 (41.7%) ($p$ = 0.021). Children presented with abdominal pain were more diagnosed with UTI, 81 (52.9%), than those without abdominal pain, 25 (11.6%) ($p$ < 0.001). The proportion of UTI was found to be higher among children who used nappies, 56 (34.8%), than in children who did not use nappies, 50 (24.2%) ($p$ = 0.026). Uncircumcised male children had more UTI, 55 (33.3%) than circumcised children, 3 (10.3%) ($p$ = 0.030). The proportion of UTI was significantly higher in children presented with vomiting 37 (44.0%) than in those without vomiting 69 (24.3%) ($p$ < 0.001). The proportion of UTI was higher in children with a body temperature above 39˚c, 11 (55.0%) than in children with a body temperature between 38–39˚C, 95 (27.3%) ($p$ = 0.008). There was no difference in UTI proportion based on sex, history of recurrent UTI, place of residence, nutrition status, or chronic disease, and the p-value was greater than 0.05 (Table 1).

**Table 1. Proportion of UTI among the febrile under-five children attending Magu District Hospital in Mwanza region, March to April 2023.**

| Variable | Description | Frequency N (%) | UTI negative n (%) | UTI positive n (%) | P value |
|---|---|---|---|---|---|
| **Age group (months)** | <24 | 151(41.0) | 98 (64.9) | 53 (35.1) | 0.026 |
| | ≥24 | 217(59.0) | 164 (75.6) | 53 (24.4) | |
| **Sex** | Male | 194 (52.7) | 136 (70.1) | 58 (29.9) | 0.625 |
| | Female | 174 (47.3) | 126 (72.4) | 48 (27.6) | |
| **Place of residence** | Urban | 72 (19.6) | 52 (72.6) | 20 (27.8) | 0.830 |
| | Rural | 296 (80.4) | 210 (70.9) | 86 (29.1) | |
| **Duration of fever (days)** | ≤2 | 155 (42.1) | 122 (78.8) | 33 (21.3) | 0.021 |
| | 3–7 | 201 (54.6) | 133 (66.2) | 68 (33.8) | |
| | >7 | 12 (3.3) | 7 (58.3) | 5 (41.7) | |
| **Axillary Temperature** | 38–39 | 348 (94.6) | 253 (72.7) | 95 (27.3) | 0.008 |
| | >39 | 20 (5.4) | 9 (45) | 11 (55.0) | |
| **Dysuria** | Yes | 61 (16.6) | 31 (50.8) | 30 (49.2) | < 0.001 |
| | No | 307 (83.4) | 231 (75.2) | 76 (24.8) | |
| **Vomiting** | Yes | 84 (22.8) | 47 (56) | 37 (44.0) | < 0.001 |
| | No | 284 (77.2) | 215 (75.7) | 69 (24.3) | |
| **Abdominal pain** | Yes | 153 (41.6) | 72 (47.1) | 81 (52.9) | < 0.001 |
| | No | 215 (58.4) | 190 (88.4) | 25 (11.6) | |
| **Recurrent UTI** | Yes | 24 (6.6) | 17 (70.8) | 7 (29.2) | 0.968 |
| | No | 344 (93.5) | 245 (71.2) | 99 (28.8) | |
| **Use of nappy** | Yes | 161 (43.8) | 105 (65.2) | 56 (34.8) | 0.026 |
| | No | 207 (56.2) | 157 (75.8) | 50 (24.2) | |
| **Chronic disease** | Yes | 8 (2.2) | 6 (75) | 2 (25.0) | 0.810 |
| | No | 360 (97.8) | 256 (71.1) | 104 (28.9) | |
| **Nutrition status** | Malnourished | 27 (7.3) | 17 (63) | 10 (37.0) | 0.326 |
| | Well-nourished | 341 (92.7) | 245 (71.8) | 96 (28.2) | |
| **Male circumcision (n = 194)** | Yes | 29 (14.9) | 26 (89.7) | 3 (10.3) | 0.030 |
| | No | 165 (85.1) | 110 (66.7) | 55 (33.3) | |

UTI—Urinary Tract Infection.

## Uropathogens bacteria causing UTI

One hundred six pathogenic bacteria strains were isolated from 106 urine specimens of febrile under-five children. No multiple infection was obtained with significant growth of bacteria counts of ≥$10^5$ CFU/ml. Gram-negative bacteria constituted 80 (75.48%), while gram-positive bacteria counted 26 (24.52%). *E. coli* was the most common isolated pathogen, 37 (35%), followed by *S. aureus* 26/106 (25%) (Table 2).

## Factors associated with urinary tract infection

In univariate analysis, age, vomiting, dysuria, body temperature, abdominal pain, use of nappy, and uncircumcision were associated with UTI. Children below 24 months (CPR 1.4; 95% CI: 1.0–1.9, $P = 0.03$) who used nappies (CPR 1.4; 95% CI: 1.0–1.9, $P = 0.026$) and those presented with vomiting (CPR 1.8; 95% CI: 1.3–2.5, $P < 0.001$), dysuria (CPR 1.9; 95% CI: 1.4–2.7, $P < 0.001$), and body temperature above 39˚C (CPR 2.0; 95% CI: 1.4–2.7, $P = 0.02$) were more likely to have UTI than their counterparts, with the prevalence ranging between 1.4 and 2.0, $p < 0.05$. Children with abdominal pain had 4.6 times more UTI than those without

**Table 2. Frequency of bacteria isolates among children under five years attending Magu District Hospital in Mwanza region, March to April 2023.**

| Isolated bacteria | Frequency | Percentage |
|---|---|---|
| *E. coli* | 37 | 34.9 |
| *S. aureus* | 26 | 24.5 |
| *Enterobacter* spp | 10 | 9.4 |
| *Klebsiella* spp | 12 | 11.3 |
| *Acinetobacter* spp | 5 | 4.7 |
| *M. morganii* | 5 | 4.7 |
| *Proteus* spp | 4 | 3.8 |
| *P. aureginosa* | 4 | 3.8 |
| *Providencia* spp | 2 | 1.9 |
| *Citrobacter* spp | 1 | 0.9 |

(CPR 4.6; 95%CI:3.1–6.8, *p* < 0.001). Uncircumcised children had 3.2 times more chances of acquiring UTI than circumcised (CPR 3.2; 95% CI: 1.1–9.6, *P* = 0.035). Sex, duration of fever, history of recurrent UTI, place of residence, nutrition status, and a history of having chronic disease were not associated with UTI.

Eight factors were included in the multivariable analysis. Age below 24 months, abdominal pain, vomiting, dysuria, body temperature above 39˚C, use of nappy, and uncircumcision were independently associated with UTI. Children below 24 months had 1.5 times the prevalence of acquiring UTI compared to those aged 24 months and above (APR = 1.5, 95% CI = 1.1–2.1, *P* = 0.010). Furthermore, children with abdominal pain had 3.9 times the prevalence of having a UTI than those without abdominal pain (APR = 3.9, 95% CI = 2.6–5.9, *P* < 0.001). Having vomiting, dysuria, or a body temperature above 39˚C increased the prevalence of having UTI by 1.6 to 1.7 times. Using a nappy increased the prevalence of acquiring UTI by 1.2 times more than not using a nappy (APR 1.2; 95% CI: 0.6–2.4, *P* = 0.003). Uncircumcised children had 2.5 times the prevalence of acquiring UTI than circumcised (APR 2.5; 95% CI: 0.4–8.2, *P* = 0.002) (Table 3).

## Antimicrobial susceptibility patterns of isolated bacteria

*E. coli* were susceptible to cefepime, piperacillin-tazobactam, nitrofurantoin, and meropenem, ranging from 81.1% to 97.3%, while least susceptible to trimethoprim-sulfamethoxazole (37.8%), gentamycin (32.4%), and ampicillin (18.9%). *S. aureus* was susceptible to nitrofurantoin (96.2%) and ciprofloxacin (92.3%) and least susceptible to penicillin (11.5%). Susceptibility to ciprofloxacin ranges from 40% (*Acinetobacter* spp) to 100% *P. aeruginosa*, *Citrobacter* spp, *Morganella morganii (M. morganii)*, *Proteus* spp, and *Providencia* spp. Susceptibility to meropenem ranges from 20% *Acinetobacter* to 100% *Enterobacter* species. Trimethoprim-sulfamethoxazole susceptibility ranges from (37.8% *E. coli)* to 70% (*S. aureus)* (Table 4).

## Multi-drug resistant bacterial strains causing UTI

MDR was observed in 35/106 (33.0%) isolated organisms. A high rate of MDR was observed in *S. aureus* 11 (31.4%), *E. coli* 7(20.0%), and *Klebsiella* spp (20.0%). A high MDR above 50% was observed in combination of ampicillin, gentamycin, and trimethoprim-sulfamethoxazole (Table 5).

**Table 3. Factors associated with UTI among febrile children under five years in Magu District -Mwanza region, March to April 2023.**

| Variable | UTI negative n (%) | UTI positive n (%) | Bivariate analysis | | Multivariate analysis | |
|---|---|---|---|---|---|---|
| | | | CPR (95% CI) | *P* value | APR (95%CI) | *P* value |
| **Age group (months)** | | | | | | |
| <24 | 98 (64.9) | 53 (35.1) | 1.4 (1.0–1.9) | 0.030 | 1.5 (1.1–2.1) | 0.010 |
| ≥24 | 164(75.6) | 53 (24.4) | 1 | | | |
| **Sex** | | | | | | |
| Male | 136 (70.1) | 58 (29.9) | 1.0 (0.8–1.5) | 0.625 | | |
| Female | 126 (72.4) | 48 (27.6) | 1 | | | |
| **Place of residence** | | | | | | |
| Urban | 72 (72.2) | 20 (27.8) | 1 | | | |
| Rural | 296 (78.9) | 86 (29.1) | 1.0 (0.7–1.6) | 0.831 | | |
| **Duration of fever (days)** | | | | | | |
| ≤2 | 122 (78.8) | 33 (21.3) | 1 | | | |
| 3–7 | 133 (66.2) | 68 (33.8) | 1.5 (1.1–2.1) | 0.020 | 1.4 (0.9–1.9) | 0.057 |
| >7 | 7 (58.3) | 5 (41.7) | 1.5 (0.7–2.9) | 0.275 | 1.5 (0.7–2.9) | 0.179 |
| **Temperature** | | | | | | |
| 38–39 | 253 (72.7) | 95 (27.3) | 1 | | | |
| >39 | 9 (45.0) | 11 (55.0) | 2.0 (1.3–3.1) | 0.002 | 1.7 (1.1–2.8) | 0.020 |
| **Dysuria** | | | | | | |
| Yes | 31 (50.8) | 30 (49.2) | 1.9 (1.4–2.7) | < 0.001 | 1.6 (1.2–2.3) | 0.010 |
| No | 231 (75.2) | 76 (24.8) | 1 | | | |
| **Vomiting** | | | | | | |
| Yes | 47 (56.0) | 37 (44.0) | 1.8 (1.3–2.5) | < 0.001 | 1.6 (1.2–2.2) | 0.002 |
| No | 215 (75.7) | 69 (24.3) | 1 | | | |
| **Abdominal pain** | | | | | | |
| Yes | 72 (47.1) | 81 (52.9) | 4.6 (3.1–6.8) | < 0.001 | 3.9 (2.6–5.9) | < 0.001 |
| No | 190 (88.4) | 25 (11.6) | 1 | | | |
| **Recurrent UTI** | | | | | | |
| Yes | 17 (70.8) | 7 (29.2) | 1.0 (0.5–1.9) | 0.968 | | |
| No | 245 (71.2) | 99 (28.8) | 1 | | | |
| **Use of nappy** | | | | | | |
| Yes | 105 (65.2) | 56 (34.8) | 1.4 (1.0–1.9) | 0.026 | 1.2 (0.6–2.4) | 0.003 |
| No | 157 (75.8) | 50 (24.2) | 1 | | | |
| **Chronic disease** | | | | | | |
| Yes | 6 (75) | 2 (25.0) | 0.9 (0.3–2.9) | 0.815 | | |
| No | 256 (71.1) | 104 (28.9) | 1 | | | |
| **Nutrition status** | | | | | | |
| Malnourished | 17 (63) | 10 (37.0) | 1.3 (0.8–2.2) | 0.302 | | |
| Well-nourished | 245 (71.8) | 96 (28.2) | 1 | | | |
| **Male circumcision** (n = 194) | | | | | | |
| Yes | 26 (89.7) | 3 (10.3) | 1 | | | |
| No | 110 (66.7) | 55 (33.3) | 3.2 (1.1–9.6) | 0.035 | 2.5 (0.4–8.2) | 0.002 |

UTI—urinary tract infection, CPR—Crude Prevalence Ratio, APR—Adjusted Prevalence Ratio.

**Table 4. Antimicrobial susceptibility patterns of bacteria causing UTI among children under five years attending Magu District Hospital in Mwanza region, March to April 2023.**

| Bacteria isolates | N | Pattern | NIT | CIP | SXT | GEN | AMP | AMC | CTX | CRO | CAZ | MRP | FEF | TZP | AK | CL |
|---|---|---|---|---|---|---|---|---|---|---|---|---|---|---|---|---|
| E. coli | 37 | S | 32 (86.5) | 28 (75.7) | 14 (37.8) | 12 (32.4) | 7(18.9) | 25 (67.6) | 25 (67.5) | 29 (78.4) | 27 (72.9) | 36(97.3) | 30 (81.1) | 32 (86.5) | 28 (75.7) | NA |
| | | I | 2(5.4) | 5(13.5) | 8(21.6) | 1(2.7) | 13 (35.1) | 6(16.2) | 8(21.6) | 4(10.8) | 4(10.8) | 0(0) | 4(10.8) | 4(10.8) | 6(16.2) | NA |
| | | R | 3(8.1) | 4(10.8) | 15 (40.5) | 24 (64.9) | 17 (45.9) | 6(16.2) | 4(10.8) | 4(10.8) | 6(16.2) | 1(2.7) | 3(8.1) | 1(2.7) | 3(8.1) | NA |
| Klebsiella spp | 12 | S | 11 (91.7) | 10 (83.3) | 8(66.7) | 3(25.0) | (0.0) | 7(58.3) | 6(50.0) | 7(58.4) | 8(66.7) | 11(91.7) | 10 (83.3) | 10 (83.3) | 8(66.7) | NA |
| | | I | 1(8.3) | 1(8.3) | 0(0) | 3(25.0) | 2(16.7) | 3(25.0) | 3(25.0) | 1(8.3) | 0(0) | 0(0) | 1(8.3) | 1(8.3) | 0(0) | NA |
| | | R | 0(0) | 1(8.3) | 4(33.3) | 6(50.0) | 10 (83.3) | 2(16.7) | 3(25.0) | 4(33.3) | 4(33.3) | 1(8.3) | 0(0) | 1(8.3) | 4(33.3) | NA |
| Enterobacter spp | 10 | S | 8(80.0) | 9(90.0) | 6(60.0) | 3(30.0) | 3(30.0) | 10 (90.0) | 6(60.0) | 7(70.0) | 9(90.0) | 10 (100.0) | 8(80.0) | 9(90.0) | 9(90.0) | NA |
| | | I | 1(10.0) | 1(10.0) | 1(10.0) | 0() | 1(10.0) | 1(10.0) | 4(40.0) | 2(20.0) | 0(0) | 0(0) | 2(20.0) | 0(0) | 0(0) | NA |
| | | R | 1(10.0) | 0() | 3(30.0) | 7(70.0) | 6(60.0) | 0(0) | 0(0) | 1(10.0) | 1(10.0) | 0(0) | 0(0) | 1(10.0) | 1(10.0) | NA |
| P. aeruginosa | 4 | S | NA | 4 (100.0) | NA | 0(0.0) | NA | NA | NA | NA | 1(25.0) | 4(100.0) | 4 (100.0) | 4 (100.0) | 4 (100.0) | NA |
| | | I | NA | 0(0.0) | NA | 0(0.0) | NA | NA | NA | NA | 2(50.0) | 0(0.0) | 0(0.0) | 0(0.0) | 0(0.0) | NA |
| | | R | NA | 0(0.0) | NA | 4 (100.0) | NA | NA | NA | NA | 1(25.0) | 0(0.0) | 0(0.0) | 0(0.0) | 0(0.0) | NA |
| Acinetobacter spp | 5 | S | NA | 5 (100.0) | NA | 1(20.0) | NA | NA | NA | NA | 1(20.0) | 5(100.0) | 5 (100.0) | 4(80.0) | 4(80.0) | NA |
| | | I | NA | 0(0) | NA | 0(0) | NA | NA | NA | NA | 1(20.0) | 0(0) | 0(0) | 1(20.0) | 0(0) | NA |
| | | R | NA | 0(0) | NA | 4(80.0) | NA | NA | NA | NA | 3(60.0) | 0(0) | 0(0) | 0(0) | 1(20.0) | NA |
| M. morganii | 5 | S | 3(60.0) | 5(100) | 4(80.0) | 2(40.0) | 1(20.0) | 3(60.0) | 3(60.0) | 3(60.0) | 4(80.0) | 4(80.0) | 5 (100.0) | 5 (100.0) | 5 (100.0) | NA |
| | | I | 1(20) | 0(0) | 0(0) | 0(0) | 0(0) | 1(20) | 1(20) | 0(0) | 0(0) | 1(20) | 0(0) | 0(0) | 0(0) | NA |
| | | R | 1(20) | 0(0) | 1(20) | 3(60) | 4(80) | 1(20) | 1(20) | 2(40.0) | 1(20) | 0(0) | 0(0) | 0(0) | 0(0) | NA |
| Proteus spp | 4 | S | 3(75.0) | 4 (100.0) | 3(75.0) | 1(25.0) | 0.0 | 3(75.0) | 2(50.0) | 3(75.0) | 4 (100.0) | 3(75.0) | 4 (100.0) | 4 (100.0) | 3(75.0) | NA |
| | | I | 0(0) | 0(0) | 0(0) | 0(0) | 0(0) | 0(0) | 1(25.0) | 0(0) | 0(0) | 0(0) | 0(0) | 0(0) | 0(0) | NA |
| | | R | 1(25.0) | 0(0) | 1(25.0) | 3(75.0) | 4(100) | 1(25.0) | 1(25.0) | 1(25.0) | 0(0) | 1(25.0) | 0(0) | 0(0) | 1(25.0) | NA |
| Other Enterobacterales | 3 | S | 3 (100.0) | 3 (100.0) | 0(0) | 1(33.3) | 0(0.0) | 2(66.7) | (66.7) | (66.7) | (100.0) | (100.0) | (100.0) | (100.O) | (100.0) | NA |
| | | I | 0(0) | 0(0) | 2(66.7) | 1(33.3) | 0(0) | 0(0) | 0(0) | 1(33.3) | 0(0) | 0(0) | 0(0) | 0(0) | 0(0) | NA |
| | | R | 0(0) | 0(0) | 1(33.3) | 1(33.3) | 3 (100.0) | 1(33.3) | 1(33.3) | 0(0) | 0(0) | 0(0) | 0(0) | 0(0) | 0(0) | NA |
| S. aureus | 26 | S | 25 (96.2) | 24 (92.3) | 1(3.8) | 19 (73.1) | NA | NA | NA | NA | NA | NA | NA | NA | NA | 20 (76.9) |
| | | I | 0(0) | 0(0) | 2 (7.7) | 0(0.0) | NA | NA | NA | NA | NA | NA | NA | NA | NA | 0(0) |
| | | R | 1(3.8) | 2(7.7) | 22 (84.5) | 7(26.9) | NA | NA | NA | NA | NA | NA | NA | NA | NA | 6(23.1) |

Keynote: NIT-nitrofurantoin, CIP-ciprofloxacin, SXT-trimethoprim-sulfamethoxazole, GEN-gentamycin, AMP-ampicillin, AMC- amoxicillin clavulanic acid, CXT-cefotaxime, CRO-ceftriaxone, CAZ-ceftazidime, MRP- meropenem, FEF-cefepime, AK-amikacin, TZP-piperacillin tazobactam, CL-clindamycin, NA-not applicable, other bacteria spp-Providencia spp (2), and Citrobacter spp (1).

**Table 5. Patterns of MDR bacteria among children under five years attending Magu District Hospital in Mwanza region, March to April 2023.**

| Antimicrobial agent | n (%) | *Acinetobacter* spp | *E. coli* | *Enterobacter* spp | *Klebsiella* spp | *S. aureus* | *P. aeruginosa* | *Proteus* spp | *M. Morganii* |
|---|---|---|---|---|---|---|---|---|---|
| AMP, GEN, SXT | 18 (51.4) | 2 | 3 | 1 | 3 | 8 | 0 | 0 | 1 |
| AMP, GEN, CIP | 5 (14.3) | 0 | 2 | 0 | 0 | 3 | 0 | 0 | 0 |
| AMP, CIP, NI, AMC | 3 (8.6) | 1 | 0 | 0 | 1 | 0 | 0 | 1 | 0 |
| AMP, CIP, MERO | 2 (5.7) | 0 | 1 | 0 | 1 | 0 | 0 | 0 | 0 |
| AMP, GEN, CIP, NI | 2 (5.7) | 1 | 0 | 0 | 1 | 0 | 0 | 0 | 0 |
| TZP, GEN, CAZ | 5 (14.3) | 0 | 1 | 0 | 1 | 0 | 3 | 0 | 0 |
| **MDR, n (%)** | 35/106 (33.0) | 4 (11.4) | 7 (20.0) | 1 (2.9) | 7 (20.0) | 11 (31.4) | 3 (8.6) | 1 (2.9) | 1 (2.9) |

Keynote: AMP-ampicillin, GEN-gentamycin, SXT-trimethoprim-sulfamethoxazole, CIP-ciprofloxacin, NIT-nitrofurantoin, MERO-meropenem, TZP-piperacillin tazobactam, CAZ-ceftazidime, AMC-amoxicillin clavulanic acid.

## The proportion of ESBL producers and MRSA strains

The MRSA was detected in 6/26 (23.1%) of the *S. aureus*. ESBL production was detected in 19/76 (25%) of gram-negative bacteria. Predominant ESBL-producing isolates were *E. coli*, *Klebsiella spp* and *Acinetobacter spp* spp (Fig 1).

## Discussions

The study determined the proportion of UTI, bacterial aetiology, antimicrobial susceptibility patterns, and factors associated with UTI among under-five children. Our study revealed that around 29% of under-five children with signs and symptoms of UTI seeking health care service at the district hospital had culture-confirmed UTI. Several pathogens were identified as the causative agents of UTI, about three-quarters being gram-negative bacteria. *E. coli* was the most isolated pathogen, constituting 35% of all pathogens, followed by *S. aureus*, constituting

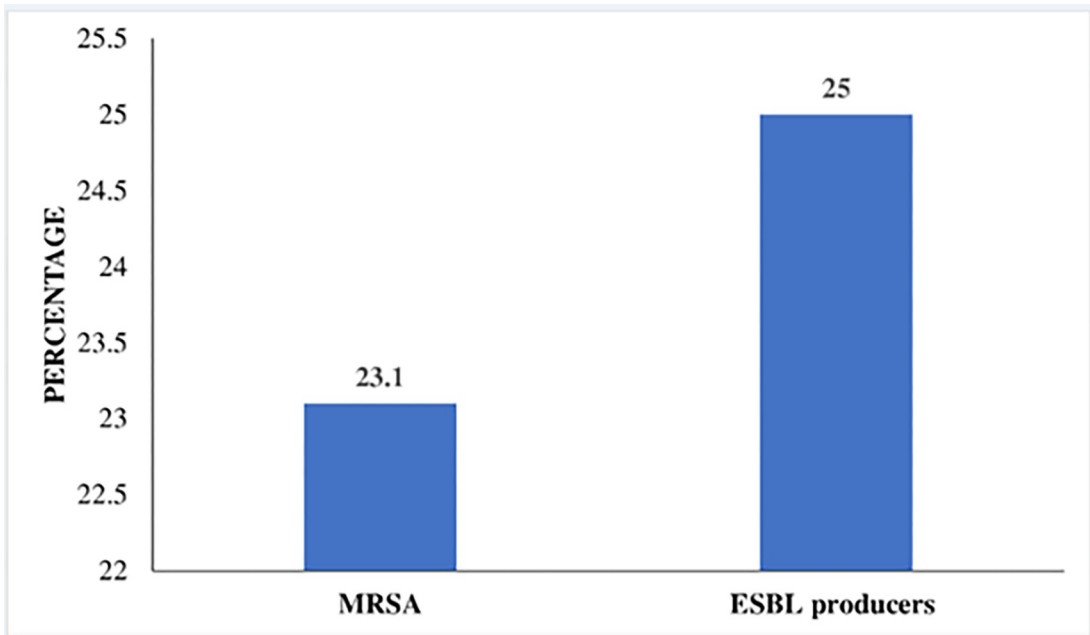

**Fig 1. ESBL producers and MRSA isolates recovered from the urine culture specimens of the febrile under-five children attending Magu District Hospital in Mwanza region, March to April 2023.**

25%. The isolated pathogens were least susceptible to trimethoprim-sulfamethoxazole, genta-mycin, penicillin and ampicillin, ranging from 11% to 38%. We found that UTI was significantly higher in children below 24 months, those presented with fever above 39˚C, abdominal pain, dysuria, and those who used nappy and uncircumcised male children.

The proportion of culture-confirmed UTI observed in our study was lower than that of a study conducted among under-fives in a tertiary hospital in the same region, which reported a proportion of 39.7% [28]. However, our study reports a higher proportion of UTI than a study conducted in Kenya (11.9%) in young children in rural areas [29]. In other studies conducted in Tanzania, the proportion of UTI ranged from 11% to 17% [6,9,24]. The variation of culture-confirmed UTI between the current study and previous studies among under-five children can be explained by geographical and facility-level differences. The differences in risk factors, such as low hygienic standards and malnutrition in different populations, contribute to the variations in the proportion of UTIs [11]. Additionally, only children with signs and symptoms of UTI were included in this study, which may account for the higher prevalence than that conducted at Kilimanjaro, which included children with and without symptoms of UTI [24].

The common causative agents of UTI in under-five children revealed in the current study were *E. coli*, followed by *S. aureus*. Our findings were consistent with several studies where *E. coli* was the predominant pathogen, with a slight variation on the second common pathogen [3,8,9]. For example, a study at a tertiary hospital in Dar es Salaam, Tanzania, reported *E. coli* as the common isolate, followed by *Klebsiella* spp, then *S. aureus* [15]. The bacterial spectrum obtained in the current study is nearly identical to the study conducted among children in Kilimanjaro, Tanzania, which identified *E. coli*, *S. aureus*, *Klebsiella* spp, *Proteus*, and *P. aeruginosa* as the common aetiologies of UTI in children [24].

Most isolates in our study were sensitive to nitrofurantoin, ciprofloxacin, meropenem, cefepime, and piperacillin-tazobactam, ranging from 80% to 100%. Our results on the pathogen's susceptibility patterns to the five antibiotics (nitrofurantoin, ciprofloxacin, meropenem, cefepime, and piperacillin-tazobactam) agree with similar studies conducted in different locations in Tanzania [9,15,24] and Nigeria [1]. The high susceptibility of isolated bacteria to most of these antibiotics, especially meropenem, may be attributed to the fact that injectable beta-lactam antibiotics are not frequently available in Tanzanian pharmacies. As a result, meropenem is less routinely administered in Tanzania than other more widely available antibiotics. Ciprofloxacin, on the other hand, is not commonly used in children under the age of 12 years. In addition, nitrofurantoin is not recommended for use in infants younger than six weeks; instead, a first-generation cephalosporin such as cephalexin may be given until the infant reaches six weeks of age.

The antibiotics frequently prescribed for empirical treatment, including trimethoprim-sulfamethoxazole, piperacillin, ampicillin, gentamycin, and penicillin, demonstrated increased resistance. These findings agree with previous studies conducted in different health facilities in Tanzania [6,9] and Kenya [29]. In contrast, a study conducted at a tertiary hospital in Dar es Salaam, Tanzania, in 2013 showed that most *E. coli* were sensitive to gentamycin [15]. The disparity in resistance rates may be attributed to the fact that these antibiotics are readily accessible at the primary healthcare level and are, therefore, being given empirically based on clinical features, accelerating antimicrobial resistance. The results of this study highlight the significance of performing a urine culture and an antimicrobial susceptibility test on febrile children.

In this study, the overall prevalence of ESBL-producing bacteria was comparatively lower than the 37% reported in a previous study conducted in the same region among malnourished under-five children admitted at a tertiary hospital [30]. Another study in Dar es Salaam, Tanzania, reported a high prevalence of faecal carriage of ESBL-producing Enterobacteriaceae (34.3%) among children [22]. Furthermore, the study identified the most predominant ESBL-

producing isolates as *E. coli*, followed by *Klebsiella* spp and *Acinetobacter* spp. This pattern is consistent with findings from other studies conducted in Ethiopia [18]. The proportion of MRSA was also lower than in previous studies conducted in India, which reported 46% among *S. aureus* from clinical isolates at tertiary hospitals [31]. The variation of rates of ESBL and MRSA in different studies indicates evidence that multidrug-resistant pathogens are more found at tertiary-level facilities than at lower health facilities. In addition, the differences in geographical locations can explain the lower proportions we found in our study.

The current study found that children under 24 months had a higher risk of acquiring a UTI than those older than 24 months. Similar findings were reported by previous studies conducted in Tanzania [9,24]. The finding may be explained by the fact that children under the age of 24 months are at risk of acquiring UTI due to underdeveloped immune systems, wearing of nappies for children under two years, and urinary tract abnormalities [11]. This study also found that the risk of UTI was 2.5 times higher in non-circumcised children compared to circumcised male children. The risk of uncircumcision for UTI is consistent with previous meta-analysis study of the risk factors for urinary tract infection in children [10]. A study conducted in Indonesia reported that circumcision was associated with a decreased incidence of UTI in children [32]. Additionally, a study conducted in Ethiopia found that uncircumcised male children have a higher risk of acquiring UTI compared to those who are circumcised [13].

This study revealed that the presence of symptoms of vomiting, difficulty urination, body temperature greater or equal to 39 degrees centigrade, and abdominal pain was associated with UTI. Our findings align with the previous studies conducted in Mwanza [9] and Nigeria [33,34]. Similar study populations, risk factors, and environmental factors may have contributed to the consistency of these findings. This study also reported that a history of recurrent UTI, chronic disease, nutrition status, and duration of fever did not show a significant association with UTI in children under five years, which agrees with other reports from Nigeria [1], England [3], and Kilimanjaro-Tanzania [24].

## Limitations of the study

Due to challenges in recall bias, we could not ascertain the prevalence of UTI and assess antimicrobial susceptibility patterns among children who had used antibiotics in the past two weeks.

## Conclusions

We observed a relatively high proportion of under-five children with culture-confirmed UTI, mainly due to *E. coli* and *S. aureus*. Several factors, including the presence of vomiting, dysuria, abdominal pain, nappy use, and uncircumcision, independently predict UTI in children. The isolated organisms showed high resistance to (trimethoprim-sulfamethoxazole, gentamycin, ampicillin, and penicillin) commonly prescribed antibiotics for treating UTI. This emphasizes the need for continuous epidemiologic surveillance in primary healthcare facilities in Tanzania. In addition, a thorough clinical analysis and periodic reviews of empirical treatment for UTI should be done based on the prevailing antibiotic susceptibility patterns.

## Supporting information

**S1 File.**
(XLSX)

## Acknowledgments

We want to convey our sincere gratitude to the Management of Sekou-Toure Regional Referral Hospital and Magu District Hospital for allowing us to conduct our research under their purview. We thank Mr Castory Dome, Mr Philipo Shokolo, and Mr Leoncy Kapinga for their technical assistance and directives during data and specimen collection. We express our gratitude to our study respondents who willingly participated and cooperated fully, thus facilitating the successful completion of this study.

## Author Contributions

**Conceptualization:** Roza Ernest, Nsiande Lema, Mtebe Majigo.

**Data curation:** Roza Ernest.

**Formal analysis:** Roza Ernest.

**Funding acquisition:** Roza Ernest, Nsiande Lema.

**Investigation:** Roza Ernest, Nsiande Lema, Mtebe Majigo.

**Methodology:** Roza Ernest, Nsiande Lema, Mtebe Majigo.

**Resources:** Roza Ernest.

**Software:** Roza Ernest.

**Supervision:** Nsiande Lema, Sued Yassin, Agricola Joachim, Mtebe Majigo.

**Validation:** Roza Ernest, Nsiande Lema, Sued Yassin, Agricola Joachim, Mtebe Majigo.

**Visualization:** Roza Ernest, Nsiande Lema, Sued Yassin, Agricola Joachim, Mtebe Majigo.

**Writing – original draft:** Roza Ernest, Nsiande Lema, Mtebe Majigo.

**Writing – review & editing:** Roza Ernest, Nsiande Lema, Sued Yassin, Agricola Joachim, Mtebe Majigo.

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
