## [Decision Letter · Decision Letter 0]

14 Nov 2023

PONE-D-23-32229Etiologic Profile, Antimicrobial Susceptibility Patterns, and Factors Associated with Urinary Tract Infection among Under-five Children at Primary Health Facility, North-Western TanzaniaPLOS ONE

Dear Dr. Ernest,

Thank you for submitting your manuscript to PLOS ONE. After careful consideration, we feel that it has merit but does not fully meet PLOS ONE’s publication criteria as it currently stands. Therefore, we invite you to submit a revised version of the manuscript that addresses the points raised during the review process.

We look forward to receiving your revised manuscript.

Kind regards,

Mengistu Hailemariam Zenebe, PhD

Academic Editor

PLOS ONE

Journal Requirements:

**Additional Editor Comments:**

Dear author, please attain the given very important comments as much as possible.

Please respond to the reviewer comments and comeback with advanced version for consideration.

Reviewers' comments:

Reviewer's Responses to Questions

**Comments to the Author**

1. Is the manuscript technically sound, and do the data support the conclusions?

Reviewer #1: Yes

Reviewer #2: Yes

2. Has the statistical analysis been performed appropriately and rigorously? 

Reviewer #1: Yes

Reviewer #2: Yes

3. Have the authors made all data underlying the findings in their manuscript fully available?

Reviewer #1: No

Reviewer #2: Yes

4. Is the manuscript presented in an intelligible fashion and written in standard English?

Reviewer #1: Yes

Reviewer #2: Yes

5. Review Comments to the Author

Reviewer #1: Thank you for inviting me to review a manuscript entitled “Etiologic Profile, Antimicrobial Susceptibility Patterns, and Factors Associated with Urinary Tract Infection among Under-five Children at Primary Health Facility, North-3Western Tanzania” Primarily, authors tried to determine culture confirmed UTI among febrile children at health care setting. They clearly presented justification to conduct the present study and they also discussed relevant literatures in similar topic both in the introduction and discussion section of the manuscript. To complement the manuscript, I listed my comments below

Major comments

1. I am not sure how UTI often leads to septicemia, dehydration. Sometimes from kidney infection bacteria may have chance to enter blood stream and cause bacteremia. Line #47-48 “Urinary tract infection (UTI) is common in under-five children presenting with fever with

significant consequences leading to septicemia, dehydration, kidney scarring…”

2. I suggest authors to describe the source and study population including Eligibility criteria for selection of the study participants. In this study it was mentioned that under five children with fever who visited the health care setting. Fever can indicate server condition (Acute febrile illness). Were there any additional criteria to select? What other laboratory tests were performed? Were infant included?

3. Was there multiple infections?

4. The description of AST in the manuscript may need revision example ‘highly susceptible’ as far as I know interpretation of AST result is Resistant, Intermediate or susceptible based on cut point. “E. coli were highly susceptible to cefepime”

5. Was the tool (questionnaire) used in the study validate? Briefly describe

6. It was indicated that the urine specimens were stored at 4oC until they tested. For how long were they stored? How far regional lab is from the hospital where study participants were recruited. The integrity of the specimen is key for recovery of bacteria.

7. Describe how antibiotics used in the study were selected

8. Did authors use only CLED for identification of Gram-positive and Gram-negative bacteria? How S. aureus and Micrococcus was differentiated? Overall the types of culture media used for AST were not indicated.

9. As the definition of multi-drug resistance may vary it advisable to include operational definition for Multi-drug resistance.

10. The sample size for was not indicted in the main method section.

11. Consider including 95% CI for culture confirmed UTI.

12. Figure 1, Y-axis is not labeled. Also include frequency and % on the top of each bar

13. It is difficult to understand the results presented in Table 3. Consider including R, I, and S

Minor comments

1. Line# 135 “…..Identify…..’ the first letter should not be capital

2. ‘Gram-positive’ and ‘gram-positive’ is interchangeably used. It is good to stick to one. ‘Gram’ or ‘gram’

Reviewer #2: Look at the comments on the sticky note on the PDF. Modify all the comments and justify for not accepted comments. I hope the manuscript would improve a lot if the comments addressed properly. The manuscript has some important data but also miss leads the readers unless otherwise properly reviewed and edited.

6. PLOS authors have the option to publish the peer review history of their article (what does this mean?). If published, this will include your full peer review and any attached files.

Reviewer #1: No

Reviewer #2: **Yes: **Tsegeye Alemayehu

---

## [Author Response · Author response to Decision Letter 0]

17 Jan 2024

Editor’s comments

Response: We have tried our best to write the manuscript following PLOS ONE’s style

2. We suggest you thoroughly copyedit your manuscript for language usage, spelling, and grammar. If you do not know anyone who can help you do this, you may wish to consider employing a professional scientific editing service. Whilst you may use any professional scientific editing service of your choice, PLOS has partnered with both American Journal Experts (AJE) and Editage to provide discounted services to PLOS authors. Both organizations have experience helping authors meet PLOS guidelines and can provide language editing, translation, manuscript formatting, and figure formatting to ensure your manuscript meets our submission guidelines. To take advantage of our partnership with AJE, visit the AJE website (http://learn.aje.com/plos/) for a 15% discount off AJE services. To take advantage of our partnership with Editage, visit the Editage website (www.editage.com) and enter referral code PLOSEDIT for a 15% discount off Editage services. If the PLOS editorial team finds any language issues in text that either AJE or Editage has edited, the service provider will re-edit the text for free. Upon resubmission, please provide the following: The name of the colleague or the details of the professional service that edited your manuscript. A copy of your manuscript showing your changes by either highlighting them or using track changes (uploaded as a *supporting information* file). A clean copy of the edited manuscript (uploaded as the new *manuscript* file)”

Response: The manuscript has been thoroughly checked for language usage, spelling and grammar using Grammarly Premium 

3. In your Data Availability statement, you have not specified where the minimal data set underlying the results described in your manuscript can be found. PLOS defines a study's minimal data set as the underlying data used to reach the conclusions drawn in the manuscript and any additional data required to replicate the reported study findings in their entirety. All PLOS journals require that the minimal data set be made fully available. For more information about our data policy, please see http://journals.plos.org/plosone/s/data-availability. Upon re-submitting your revised manuscript, please upload your study’s minimal underlying data set as either Supporting Information files or to a stable, public repository and include the relevant URLs, DOIs, or accession numbers within your revised cover letter. For a list of acceptable repositories, please see http://journals.plos.org/plosone/s/data-availability#loc-recommended-repositories. Any potentially identifying patient information must be fully anonymized. Important: If there are ethical or legal restrictions to sharing your data publicly, please explain these restrictions in detail. Please see our guidelines for more information on what we consider unacceptable restrictions to publicly sharing data: http://journals.plos.org/plosone/s/data-availability#loc-unacceptable-data-access-restrictions. Note that it is not acceptable for the authors to be the sole named individuals responsible for ensuring data access. We will update your Data Availability statement to reflect the information you provide in your cover letter.

Response: The study’s minimal underlying data set has been uploaded as a Supporting Information files

4. We note that you have indicated that data from this study are available upon request. PLOS only allows data to be available upon request if there are legal or ethical restrictions on sharing data publicly. For more information on unacceptable data access restrictions, please see http://journals.plos.org/plosone/s/data-availability#loc-unacceptable-data-access-restrictions. In your revised cover letter, please address the following prompts:

Response: The study’s minimal underlying data set has been uploaded as a Supporting Information files

Response: The study obtained ethical clearance number MUHAS-REC-12-2022-1484 from the Muhimbili University of Health and Allied Sciences Senate Research and Publication Committee. This has been addressed in lines 228 to 230.

Comments from reviewer 1

1.I am not sure how UTI often leads to septicemia, dehydration. Sometimes from kidney infection bacteria may have chance to enter blood stream and cause bacteremia. Line #47-48 “Urinary tract infection (UTI) is common in under-five children presenting with fever with significant consequences leading to septicemia, dehydration, kidney scarring”

Response: The sentence has been revised in line 55 of the introduction section.

2. I suggest authors to describe the source and study population including Eligibility criteria for selection of the study participants. In this study it was mentioned that under five children with fever who visited the health care setting. Fever can indicate server condition (Acute febrile illness). Were there any additional criteria to select? What other laboratory tests were performed? Were infant included?

Response: We involved children aged 2 months and above who met the inclusion criteria. We have to revise and write: We recruited children with fever who the attending clinician suspected to have UTI based on signs and symptoms such as fever of 38oC or above, pain or burning when urinating, frequency urination, discoloured urine, lower abdominal pain, or vomiting.. The eligibility criteria have been added; line 106 to 111

3. Was there multiple infections?

Response: No multiple infection was obtained with significant growth of bacteria counts of �10⁵ CFU/ml. We have rephrased the result section to reflect this finding line 275 to 276.

4. The description of AST in the manuscript may need revision example ‘highly susceptible’ as far as I know interpretation of AST result is Resistant, Intermediate or susceptible based on cut point. “E. coli were highly susceptible to cefepime”

Response: This has been addressed in table 4 of the manuscript, interpretation of AST which includes resistant, intermediate and sensitive were added in the table.

5. Was the tool (questionnaire) used in the study validate? Briefly describe

Response: Yes, the questionnaire used for data collection was validated, and this is explained in the data collection section, line 124.

6. It was indicated that the urine specimens were stored at 4oC until they tested. For how long were they stored? How far regional lab is from the hospital where study participants were recruited. The integrity of the specimen is key for recovery of bacteria.

Response: Samples were stored in the refrigerator for not more than 12 hours; we have indicated this in the sample collection section line 141.

7. Describe how antibiotics used in the study were selected

Response: Descriptions on how antibiotics used in the study were selected has been added in line 174 to 175. 

8. Did authors use only CLED for identification of Gram-positive and Gram-negative bacteria? How S. aureus and Micrococcus was differentiated? Overall, the types of culture media used for AST were not indicated.

Response: We differentiated Staphylococcus aureus from Micrococcus species by observing colony characteristics and performing a gram stain. The type of culture used for AST has been indicated in lines 175-177

9. As the definition of multi-drug resistance may vary it advisable to include operational definition for multi-drug resistance.

Response: We have added in the materials and methods a subsection of the definition of terms. The definition of multi-drug resistance has been included in lines 143-153

10. The sample size for was not indicted in the main method section.

Response: The sample has been indicated in methodology line 114

11. Consider including 95% CI for culture confirmed UTI.

Response: 95% CI has been added to the culture confirmed UTI in the abstract, results and discussion sections 

12. Figure 1, Y-axis is not labelled. Also include frequency and % on the top of each bar

Response: Figure one was changed to a table 2 which indicates frequency and percentage

13. It is difficult to understand the results presented in Table 3. Consider including R, I, and S

Response: Table 3 has been revised for clarity and changed to table 4 since we added another table

Minor comments

1. Line# 135 “…..Identify…..’ the first letter should not be capital

2. ‘Gram-positive’ and ‘gram-positive’ is interchangeably used. It is good to stick to one. ‘Gram’ or ‘gram’

Response: The first letter of "identify" has been written in lowercase. "Gram positive" and "gram positive" have been revised to "gram positive." However, when starting a sentence, "Gram" is written with a capital letter

Comments from Review 2

Comments for Reviewer #2 were on the sticky note on the PDF file. We have addressed all comments and modified the manuscripts

In title you said etiological profile, is it include fungi?

Response: The title has been revised to reflect the bacterial etiology; line 1

Abstrct background:Where UTI is common infection?. Avoid use of short form in abstract. Better to be consistence as per title or should modify the title accordingly. Foristance febrile underfive,antibiotic vs antimicrobial,proportion of UTI vs etiology profile

Response: We rephrased the sentences for clarity and to ensure consistency, and short-form words were omitted; lines 17-23

Abstract methodology: Why only febrile? Is fever the only criteria to be suspected as UTI for under five? Operationalize this. The method is not written well, rewrite again

Response: The materials and methods subsection in the abstract has been rewritten for clarity; Lines 25-32

Abstract results : rewrite the result very well. Like of all 28.8% had Lab, confirmed means? do you mean culture confirmed? write the organism name in italic. Is circumcision a common practice for both male and female or for only for male? is those for female taken as nor not applicable? Rewrite bacteria in italic. Is suceptibility pattern for overall? or E.. coli & S. aurues

Response: The result subsection of the abstract has been rewritten for clarity. Bacteria have been written in italics; the susceptibility patterns are for E.coli and S. aureus; Lines 34-43

Introduction

Add One paragraph for common cause of UTI from different literature.

Response: A paragraph detailing the common causes of UTI has been included; Line 62-69

what does the renal functions of the kidney mean?

Response: The sentence has been revised to reflex that, Early diagnosis is essential to preserve and maintain the proper functioning of the kidneys and prevent any further decline in their function. Line 56-57

uncircumcision in male? 

Response: Yes uncircumcission in male, The sentence has been paraphrased; Line 43&73

The prevalence of UTI among under-five children ranges from 7.4 % to 43.6% where this studies conducted

Response: The reported prevalence was indicated from Nigeria and Tanzania; Line 59-60

The risk factors for acquiring UTI in under-five children are multiple and differ with

geographical locations, settings, and seasons. Add reference 

Response: Reference number 9 has been added; Line 71

Antimicrobial resistance (AMR) has become significant concern worldwide.add reference

Response: Reference number 14 was added; Line 77

Can you generalize with your specific hospital data about the treatment? or can leads you to change treatment guidelines?

Response: The sentence has been ommited, since the same information were discussed in line 89-94

Materials and methods

Line 91, Can you clarify on primary health care and district hospital? in your title it says primary health care whereas the study is conducted in district hospital? Besides this your study area description don't indicate as whether it is urban or rural but in reasoning or gab analysis your said that rural community lacks attention?

Response: Yes, in Tanzania, a district hospital is considered a primary healthcare facility. The Magu district, where our research was conducted, is situated in the rural parts of the region. However, when classifying the demographic data, we categorized urban as individuals from the town area of Magu, and rural as those from other areas outside the town; Line 103

why only children with fever? 

Response: Fever is a common symptom of various illnesses, the aim of recruiting children with fever ensured that,study participants have a higher likelihood of having the targeted condition; Line 106-109

The total sample size should be indicated here. 

Response: The total sample size was indicated; Line 114

In Data collection your sociodemographic is limited why? 

Response: The scope and objectives of this study prioritized these sociodemographic variables; Line 121

Is Sekou Toure Regional Referral Hospital different from above hospital?

Response: Yes, the study was conducted at Magu District Hospital. However, the samples were analyzed at Sekou Toure Regional Referral Hospital since the District hospital lacked the capacity to process culture samples

Line 131- 132, write more how do you handle in case of mixed growth 

Response: Handling of mixed growth of bacteria has been indicated; Line 161-162

Line 135, Identify should start with small letter

Response: The word Identify written starting with small letter Line 165

is this the only biochemical test? how do you identify like P. aerigenosa & Acitenobacter spp 

Response: Oxidase test used to identify P. aerigenosa & Acitenobacter spp has been added Line 168-169

Antimicrobial susceptibility testing: Can we report erthromycin for organisms from UTI? this shows you haven't followed the CLSI properly. 

Response: Erthromycin disk has been removed from this manuscript

Extended-Spectrum Beta-Lactamase Detection: Is this for the three or for one? if for one you have write cefotaxime or cftriaxone or ceftazidime. is the increase of zone of inhibition towards CAZ or CXT or both?

Response: The sentence has been rephrased Line 192-193, 199-201

Quality assurance: As indicated in the quideline, on which guidelines? 

Response: The sentence” as indicated in the quideline was ommited since checking expiration date and assess reagents does not need any guideline.

Data analysis: ≤ 0.2, this cut of value most of the time is 0.25. Do you have some justification for it?

Response: The choice of a cut-off value in data analysis can depend on the specific field of study, the nature of the data, and the research question being addressed, This

---

## [Decision Letter · Decision Letter 1]

21 Feb 2024

PONE-D-23-32229R1Bacterial etiologic Profile, Antimicrobial Susceptibility Patterns, and Factors Associated with Urinary Tract Infection among Under-five Children at Primary Health Facility, North-Western TanzaniaPLOS ONE

Dear Dr. Ernest,

Thank you for submitting your manuscript to PLOS ONE. After careful consideration, we feel that it has merit but does not fully meet PLOS ONE’s publication criteria as it currently stands. Therefore, we invite you to submit a revised version of the manuscript that addresses the points raised during the review process.

Dear Author,

I have seen that you have made a major correction and your reviewers have some more correction on it. could you adjust it as needed?

We look forward to receiving your revised manuscript.

Kind regards,

Mengistu Hailemariam Zenebe, PhD

Academic Editor

PLOS ONE

Journal Requirements:

Reviewers' comments:

Reviewer's Responses to Questions

**Comments to the Author**

1. If the authors have adequately addressed your comments raised in a previous round of review and you feel that this manuscript is now acceptable for publication, you may indicate that here to bypass the “Comments to the Author” section, enter your conflict of interest statement in the “Confidential to Editor” section, and submit your "Accept" recommendation.

Reviewer #1: All comments have been addressed

Reviewer #2: All comments have been addressed

2. Is the manuscript technically sound, and do the data support the conclusions?

Reviewer #1: Yes

Reviewer #2: Yes

3. Has the statistical analysis been performed appropriately and rigorously? 

Reviewer #1: Yes

Reviewer #2: Yes

4. Have the authors made all data underlying the findings in their manuscript fully available?

Reviewer #1: Yes

Reviewer #2: Yes

5. Is the manuscript presented in an intelligible fashion and written in standard English?

Reviewer #1: Yes

Reviewer #2: Yes

6. Review Comments to the Author

Reviewer #1: Thank you for addressing most of my comments.

Comment 8 was not fully addressed I don’t think morphology and gram staining is sufficient to differentiate Staphylococcus aureus from Micrococcus you may indicate this in limitations.

Reviewer #2: Hi, dear authors!

You did a very good job of responding to the comments, and the response is fascinating. However, the manuscript still requires work in a few areas, primarily related to the discussion. Please refer to the sticky note in the PDF file that is attached. I do have a concern about some variables and associated factors, and before making a decision, these things should be resolved.

7. PLOS authors have the option to publish the peer review history of their article (what does this mean?). If published, this will include your full peer review and any attached files.

Reviewer #1: **Yes: **Musa Ali

Reviewer #2: **Yes: **Tsegaye Alemayehu

---

## [Author Response · Author response to Decision Letter 1]

10 Apr 2024

POINT TO POINT RESPONSE TO EDITOR AND REVIEWERS’ COMMENTS

Editor’s comments

Response: All references have been reviewed and errors corrected

Comments from Reviewer #1

Thank you for addressing most of my comments. Comment 8 was not fully addressed I don’t think morphology and gram staining is sufficient to differentiate Staphylococcus aureus from Micrococcus you may indicate this in limitations.

Response: Thank you for the comment. We have rephrased the paragraph related to previous comment 8 to indicate that We differentiated S. aureus from other Staphylococcus and Micrococcus species by coagulase tests. Line 164-165

Comments from Review # 2

Comments from Reviewer #2 were on the sticky note on the PDF file. We have extracted the comments from the PDF file and addressed all of them as follows:

1. Materials and methods in the abstract: 

i. Method only; remove material 

ii. Bring “District Hospital in Mwanza Region, North-Western Tanzania, between March and April 2023 to the objective to make it SMART and remove from the material and methods.

iii. Still, the statistical analysis is not clearly written. OR, and CI, cut of a p-value.

Response: We have addressed the comments related to the materials and methods in the abstract section as suggested. Line 23-30

2. Results in the abstract: 

i. Make the urinary tract infections culture-confirmed

ii. Multidrug resistance was observed in 33.0% of isolates, Methicillin Staphylococcus aureus in 23.1% of Staphylococcus aureus, and extended-spectrum beta-lactamases in 25% of gram-negative bacteria. Make this in separate sentences as they are different, and the sentence does not make sense.

Response: we have revised to indicate culture-confirmed urinary tract infection and separated the sentences as indicated for clarity. line 33-40

iii. Urinary tract infection was associated with vomiting, age, dysuria, abdominal pain, nappy use, and uncircumcision in males. Should be indicated which category associated with UTI, age, for instance, which range?, vomiting presence or absence, abdominal pain presence or absence similarly in conclusion.

Response: We have revised the sentence in the abstract result section to clearly indicate the categories associated with UTI. Line 39-40

3. Conclusion in abstract

what are commonly prescibed antibioitcs? Is your recomendation only the suspetibitly pattern? What about prevention and control based on your findings from associated factors?

Response: The conclusion has been revised to include the list of common antibiotics and recommendations regarding associated factors. Line 41-45

4. Keywords:

Add some words like ethiology agent, associated factros.

Response: We have added the words aetiologic agent and associated factors to the list of keywords. Line 49

5. Use a square bracket for all references.

Response: We have used square brackets for all references

6. Studies conducted in Nigeria and Tanzania have reported the prevalence of UTI in under-five children ranging from 7.4% to 43.6% (1,4,5). Better use some national data or meta-analysis data from Africa or WHO report. why you focused on Nigeria and Tanzania?

Response: We have included reference # 4 with a systematic review and meta-analysis data. Line 57

7. First time, write long and short forms and use short forms throughout the document.

Response: The comment has been accepted and addressed in the entire manuscript 

8. In the study design, setting, and population, the study involved under-five children with a fever seeking healthcare services. Do you think all UTI patients presented with fever? What is the standard for under-five children to be suspected of UTI? Unless otherwise, this leads to a selection bias.

Response: We have revised the paragraph for clarity. Line 101 - 104

9. We recruited children with fever who the attending clinician suspected to have UTI based on signs and symptoms such as fever of 38 degrees Celsius or above, pain or burning when urinating, frequent urination, discoloured urine, lower abdominal pain, or vomiting. this contradict with the above sentence.

Response: As per the response to comment # 8 above, we have revised the paragraph for clarity and to avoid contraction. Line 101-104

10. The study did not include children who used catheters within 72 hours because catheter insertion may directly inoculate microorganisms into the urinary bladder. reference is needed which stanard recomend this?

Response: We have added reference # 23 which indicates the effect of catheterization on UTI. Line 106

11. Risk factors: do you think nappy use works for all?

Response: Nappy use as a risk factor for UTI may not work for all but has been indicated as a factor in other study 

12. Definition of terms: Make in bullect with bold for the title. also include for UTI.

Response: We have written the sentence in the definition of terms as suggested and included the definition of UTI. Line 137-147

13. Do you collect all the specimens with midstream? What about for catheters?

Response: For the purpose of this study, we instructed parents and guardians to collect midstream urine. We did not use a catheter for urine collection. 

14. In Table 1: *Frequency and percentage of participants calculated as number within the variable category divided by the total sample size of the variable, ** proportion of patients within the variable category with no Urinary Tract Infection calculated as number of divided by the total number of individuals in the variable category. *** proportion of patients within the variable category with Urinary Tract Infection calculated as number of cases divided by the total number of individuals in the variable category. I don't think this is important, remove from table to *.

Response: The table legend has been removed as suggested. Line 260

15. In Table 4, Morganella spp. You said M. morgannii in the above makes it uniform.

Response: We have changed to M. morganii for uniformity. Line 312

16. In table 4: Give the title keynote: NIT= Nutrofurantion; it is better to use NIT-Nutrofurantion.

Response: The Table 4 legend has been corrected as suggested. Line 313-315

17. Discussion: use the figure and country; double referencing is not important. similarly correct for others. all prevalence figures from another study should be included in all the discussion. Unless otherwise it is difficult to now whether higher, lower or inline.

Response: The discussion section has been re-worked to address the comment accordingly

---

## [Editor Report · Decision Letter 2]

12 Apr 2024

Bacterial aetiology, Antimicrobial Susceptibility Patterns, and Factors Associated with Urinary Tract Infection among Under-five Children at Primary Health Facility, North-Western Tanzania

PONE-D-23-32229R2

Dear Dr. Ernest,

We’re pleased to inform you that your manuscript has been judged scientifically suitable for publication and will be formally accepted for publication once it meets all outstanding technical requirements.

Kind regards,

Mengistu Hailemariam Zenebe, PhD

Academic Editor

PLOS ONE